


# Simulation of a lithosphere-atmosphere-ionosphere electromagnetic
# coupling prior to the Wenchuan $M_S$8.0 earthquake

3       Mei Li[1,*],Zhuangkai Wang[2],Chen Zhou[2] , Handong Tan[3], Meng Cao[4]

[1]Institute of Earthquake Forecasting, China Earthquake Administration, Beijing 100036, China
[2]Department of Space Physics, School of Electronic Information, Wuhan University, Wuhan 430072, China
[3]China University of Geosciences, Beijing 100083, China
[4]Sichuan Hydropower Engineering Geophysical Exploration Co., Ltd, Chengdu 610072, China
*Correspondence to:* M. Li (mei_seis@163.com)
**Abstract** Continuously to a previous work on qualitatively investigating the probable electromagnetic
interacting process among lithosphere, atmosphere and ionosphere, this work aims to quantitatively
establish an electromagnetic coupling model among these three spheres prior to the Wenchuan
earthquake. Firstly, a underground finite length electrical dipole in a half-space model has been
employed to estimate the possible "energy source" for an observable 1.3 mV m$^{-1}$ electrical field
registered at 1440 km Gaobeidian station during the Wenchuan event. The result shows that the
seismo-telluric current covers ~$10^5$–$10^6$ kA if the measuring frequency $f$ = 0.01–10 Hz considered. The
central magnitude of the vertical electrical field caused by the current at 0.01 Hz on the Earth's surface
can be up to kV m$^{-1}$. Then, this vertical field acts as an input into an electric field penetration model. It
is shown that this field attenuates quickly at the atmosphere and completely vanishes at the top
ionosphere and produces a 0.1 mV m$^{-1}$ additional electrical field at the ionospheric bottom. Through
the TIE-GCM, this additional electrical field causes 0.01% ionospheric variations on electron density
and TEC near the Wenchuan epicenter, as well as near its magnetically conjugated point. Further, the
simulations have also been discussively performed on frequencies of 1 Hz and 10 Hz. The results
demonstrate that the variations of electron density present their maximum values at the height of ~300–
400 km and the varied percentages of ionospheric parameters have been beyond 10%, the same
magnitude as what has been registered during the Wenchuan shock.










## 1. Introduction

So far, short-term earthquake prediction is still one of the most challenging targets worldwide, but the investigations on probable earthquake precursors in the fundamental geophysical framework should be better than trying to guess the future (Eftaxias et al., 2003; Prescott, 2019; Hough, 2020; Conti et al., 2021). An earthquake (EQ) is a systematical geodynamic process that develops gradually as a strain accumulates slowly for several years, and culminates with a sudden rupture and displacement of a fault in the rigid lithosphere (Bock, 1994). Exploring and understanding for possible earthquake precursors, especially short-term ones, has been the most promising approach to short-term earthquake prediction. Electromagnetic observing on possible information originated from seismic activities is one of the most effective geophysical ways to snoop the last process of earthquake evolutionary. But there is still a controversial understanding on the issue of mechanism on producing and propagating of seismic signals from ground-based electromagnetic observation although pronounce achievements have been gained on this problem. As Earth observation from satellite develops, more and more reports have shown that the ionosphere, as a conductive part of the air, is unexpectedly sensitive to seismic activities. Ionospheric measuring has gradually shown its potential application in the field of earthquake monitoring and forecasting and emergency rescue due to its fast-speed, large-scale, and high-resolution results, especially for areas with poor natural environments. In very recent years, it has been testified that there is an energy transfer among lithosphere, atmosphere and ionosphere at the last stages of earthquake preparation. Thus, a promising way to improve the understanding on this complex geodynamic process is to integrate ground data with satellite Earth Observation combined (De Santis et al., 2015).

Investigations on seismo-ionospheric influence or lithosphere-atmosphere-ionosphere coupling (LAIC) mechanism have been primarily performed on the basis of two hypotheses: internal gravity wave (IGW) or electric field.

Gokhberg et al. (2000) have found that the irregular ionospheric variations occurred several days prior to strong earthquakes after an analysis of experimental data obtained at the last stage of earthquake preparation. These ionospheric irregularities are attributed to the propagation of IGW through the ionosphere and originated from the long wave earth oscillations, local green gas effect or an unsteady injection of lithosphere. Thus, they have supposed that the generation of IGW (acoustic gravity wave, AGW, after) should be considered as a mechanism of LAI coupling (Molchanov et al., 2004; Meister et al., 2011). However, Sorokin & Hayakawa (2013, 2014) have presented that it is difficult to interpret observing results of earthquake precursory information on the basis of IGW propagation model due to its insufficient theoretical simulations and wave-like propagating features (Conti et al., 2021). Therefore, it is easily to understand that investigations on LAI coupling transmit from an acoustic-driven mechanism to an electromagnetic coupling due to its very low effectiveness (Pulinets & Davidenko, 2014).

However, the electromagnetic "energy source" of the LAI coupling originated beneath the Earth or in the atmosphere has still been under controversial. There are generally two different viewpoints: electric sources beneath the Earth or ones on the ground. On one hand, scientists, who think that this electromagnetic source is beneath the ground, have to confront with a problem, that is the producing and propagating mechanism of electromagnetic emissions underground. Many laboratory and wild-field rock fracturing experiments have broadly been conducted to understanding the producing mechanism of electric and magnetic signals. Qian et al., (1996, 2003) have found that the large magnetic pulses with shorter periods appeared at the last stage of the experiment. These relative high


frequency signals may be induced by instantaneous electric current of the accumulated charges during
the main cracking acceleration (Hao et al., 2003). While Freund et al. (Freund & Wengeler, 1982) have
proposed that seismo-telluric current may attribute to transmission of negative charge carriers and
positive holes from stressed rocks (Freund, 2002, 2009, 2010; Freund & Sornette, 2007; Scoville et al.,
2015). Up to now, several mechanisms, like electrokinetic and magnetohydrodynamic, piezomagnetism,
and microfracturing, have been proposed to explain the producing and propagating processes of electric
and magnetic emissions observed both during seismic activity and in the laboratory experiments but no
specified one has been well established (Conti et al., 2021; Heavlin et al., 2022). To construct
underground physical or mathematics models is always being an effective way to investigate this topic
(Huang, 2011; Huang & Lin, 2010; Ren et al., 2012).
On the other hand, investigators, who believe that the "electric energy" of the LAI coupling on the
ground, think that seismic activities can trigger off radon ionization, charged particulates injection and
changes in load resistance in global electric circuit, which further leads to zonal additional electrical
field near the ground (Pulinets & Ouzounov, 2010; Pulinets & Davidenko, 2014). While Sorokin &
Hayakawa (2013, 2014) have thought that injection of charged aerosols into the atmosphere acts as
electro-motive force to cause changes of conduction current in global electric circuit, which facilities
the electrical field propagating vertically to the ionosphere. However, Pulinets & Davidenko (2014)
have reported that no scientific publications have demonstrated such injection of charged aerosols into
atmosphere before earthquakes, and the vertical external current flowing into the Global Electric
Circuit is absolutely impossible. Pulinets & Ouzounov (2010) have demonstrated that air ionization and
hydration processes induced by earthquakes in the vicinity of active tectonic faults change the global
electric circuit, which leads to a zonal additional electrical field near the Earth's surface (Pulinets &
Davidenko, 2014). Kuo et al (2011, 2014) have investigated the LAI coupling on the basis of this
p-hole theory to consider stressed seismic fault as a dynamo to drive currents from stressed rocks to the
Earth's surface. Zhou et al. (2017) have further developed an electrodynamic LAI coupling model
based on the DC electric field penetration and the results show that the LAIC electric field can
penetrate into a higher altitude in the ionosphere at a low latitude than at a high latitude. This
conclusion indicates that the additional electric field must be large enough during the LAI coupling if
expected observable plasma irregularity is obtained in the ionosphere. At the same time, an upper
atmosphere numerical model has been utilized to investigate the LAI coupling and the results have
displayed that seismogenerated zonal electric field can cause vertical plasma drift of F2 layer leading to
the disturbance of TEC (total electron content) (Namgaladze et al., 2012;Zolotov et al., 2012;Zolotov,
113  2015).
Additionally, thermal anomalies, such as ground surface latent flux and ongoing longwave
radiation (OLR), have also been considered as possible mechanism to drive LAI coupling but lacking
of a well-established model (Hayakawa & Pulinets, 2009, Pulinets et al., 2000; Liperovsky et al., 2008;
Freund, 2011). However, whatever the physical mechanism of the electromagnetic field generation is,
it seems that the electric sources or "energy sources" near the ground is undoubtedly a necessary part
during LAI coupling. Achieving a better understanding of a complex physics coupling of earthquakes
by the efforts of the involved scientific community worldwide will be dedicated to find a final answer
(De Santis et al., 2015; Conti et al. 2021).
At 14:28:01 CST (China Standard Time) on May 12, 2008, a large EQ with a magnitude of $M_S$ 8.0
hit the Wenchuan area, Sichuan province, with an epicenter located at 103.4 °E and 31.0 °N and a depth
of 19 km. This event caused major extensive damage and 69,000 people lost their lives.





As a personal experience of tracing and recording measurement in Hebei observing network
during this large event (See the Supplement), Li et al. (2009) have firstly in Chinese and then in
English (Li et al., 2013) reported remarkable visible ULF ($f$ = 0.01–10 Hz or its advantageous
frequency band $f$ = 0. 1–10 Hz) electromagnetic emissions of 1.3 mV m$^{-1}$ electrical field recorded at the
1440 km Gaobeidian observing station during the Wenchuan EQ. Utilizing an 'Earth-ionosphere'
physical model and a half-space model, Li et al. (2016) have modulated and interpreted the probability
of this abnormal phenomenon recorded at a far distance and inferred a possible seismo-elluric currents
at the depth of the Wenchuan hypocenter with and without ionospheric effect considered.
In this paper, on the basis of the work done by Li et al. (2016), this investigation mainly focuses
on the propagating process of the ground observable electrical field among the atmosphere and the
ionosphere, and the corresponding ionospheric influence caused by this field. So, in Section 2, in the
light of the observable electrical field registered during the Wenchuan EQ, we first utilize a half-space
model constructed by Li et al. (2016) to infer a probable underground seismo-elluric current, which
will act as the "energy source" driving the total LAI electromagnetic coupling. In Section 3, an electric
field penetration model developed by Zhou et al. (2017) will be employed to investigate the
propagating process of the electrical field induced by this current in the atmosphere and the ionosphere.
Additional electrical field at the bottom of the ionosphere will also calculated during this time. In
Section 4, ionospheric variations caused by this additional field will be evaluated using TIE-GCM
(Thermosphere-Ionosphere-Electrodynamics General Circulation Model) and compared with the
real-tIme ionospheric recordings during the Wenchuan event. Discussion and conclusions are in
Section 5 and Section 6, respectively.

## 2. An estimation of the LAIC "energy source" associated with the Wenchuan EQ

Li et al. (2009) have firstly presented that obvious ULF ($f$ = 0.01–10 Hz) electromagnetic
emissions were recorded at 1440 km Gaobeidian station in Hebei observing network prior to the
Wenchuan $M_S$8.0 EQ and the peak of the electrical signals reached 1.3 mV m$^{-1}$ during the climax stage
of this anomaly. Then, Li et al. (2013) have established the relative locations of the Wenchuan epicenter
and the Hebei observing networks in Figure 1 to describe the electromagnetic anomaly recorded at the
remote observing stations. Further, Li et al. (2016) had employed two physical models of a half-space
model and an "Earth-ionosphere" model to gain the probability of this unprecedented anomaly
registered by such a remote station if the ionospheric influence considered. Their results have shown
that the ionosphere can promote the electromagnetic wave propagation, which is equivalent to an
effective improvement of the detectability of the system. The system could easily recorded signals
originated from the epicentral area of seismic activities even beyond its detectable range. At that time, a
surface coordinate system had been added to Figure 1 in Li et al. (2013) to form Figure 5 in Li et al.
(2016) to comply with the models and this figure has been also employed in this paper named Figure
1a.
Here, we use Figure 1a in this paper to exhibit the relative locations of the observing stations in
Hebei ULF electromagnetic observation network and the Wenchuan epicenter, as well as a surface
coordinate system. A half-space model utilized in Li et al. (2016) has also been employed here as
Figure 1b: An x-directed dipole with a length $L$ and a current $I$ is located in the bottome medium (Earth:
$z > 0$ ). Here the Earth is considered homogeneous, with the electrical properties: magnetic
permeability $\mu_1$, permittivity $\varepsilon_1$, and conductivity $\sigma_1$. The top medium (air: z < 0) is with electrical
properties$\mu_0$, $\varepsilon_0 (= 8.854 \times 10^{-12}$ Farad m$^{-1})$ and $\sigma_0 (= 10^{-14}$ S m$^{-1})$.


The approach to investigate the electromagnetic fields emitted by a long dipole current source
follows the magnetic vector potential formulation developed by Key (2009), who developed a
generalized formulation for multiple layers below and above the transmitter. Exponential forms had
been adopted for the recursions in isotropic media, with the z axis downward. Assuming a
time-harmonic source with $e^{-i\omega t}$ time dependence, Maxwell's equations are
$$\nabla \times \mathbf{E} = i\omega \mathbf{B}, \qquad\qquad (1)$$
and
$$\nabla \times \mathbf{B} = \mu\sigma\mathbf{E} + \mu\mathbf{J}_s . \qquad\qquad (2)$$
where, magnetic permeability $\mu$ variations are negligible and angular frequency $\omega$ is low enough so
that the displacement currents can be neglected. Expression $\mathbf{J}_s = \mathbf{I}\delta(\mathbf{r} - \mathbf{r_0})$ is the imposed electric
dipole source at position $\mathbf{r_0}$ with vector moment $\mathbf{I}$, and here is limited to an infinitesimal dipole with
unit moment.

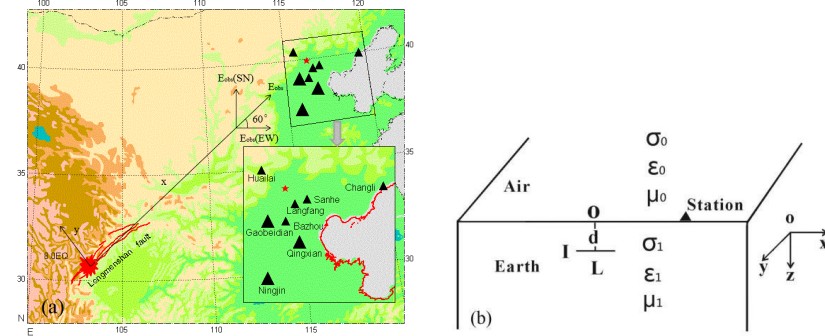

**Figure 1.** (a) Relative locations of the Wenchuan EQ epicenter and observation stations. Black solid triangles
show observing stations in Hebei observing network and bigger ones indicate the stations recording abnormal
information. The red star denotes Beijing (Li et al., 2013, Figure 1). A ground surface coordinate system has been
formed by Li et al. (2016). (b) The half-space model: an x-directed dipole current source is placed in the bottom
medium (Earth), and the dipole center coordinate is (0, 0, d).Here, z is defined positive in the downward direction.

The total space is assumed to be non-magnetic and the magnetic permeability $\mu$ variations are
negligible in the different layers, i.e. $\mu_1 = \mu_0 = 4\pi \times 10^{-7}$ Farad m$^{-1}$. On the same manner we
have $\varepsilon_1 = \varepsilon_0 = 8.854 \times 10^{-12}$ Farad m$^{-1}$, i.e. $\varepsilon_1$ is not considered as zero during all calculations.
The rupture length is set $L = 150$ km, within 30 s out of the total 90 s of the main Wenchuan rupture
(Zhang et al., 2009) and the conductivity of the Earth is $\sigma_1 = 1.0 \times 10^{-3}$ S m$^{-1}$ (Li et al., 2016)
during the calculations. For an observed 1.3 mV m$^{-1}$ electrical field at the Gaobeidian station, the
expected seismo-elluric current falls in the range of $I = 1.5 \times 10^5 - 3.4 \times 10^6$ kA for the frequency range of
$f = 0.01 - 10$ Hz. It can be seen that the current gradually increases as the observing frequency increases
due to a dramatic attenuation of the electrical fields at higher frequency. This current induces
electromagnetic emissions at the Earth's surface. Figure 2 displays two-dimensional distributions of
three electrical components of Ex, Ey and Ez, respectively, produced at the Earth's surface by the
seismo-elluric current $I = 1.5 \times 10^5$ kA when $f = 0.01$ Hz.











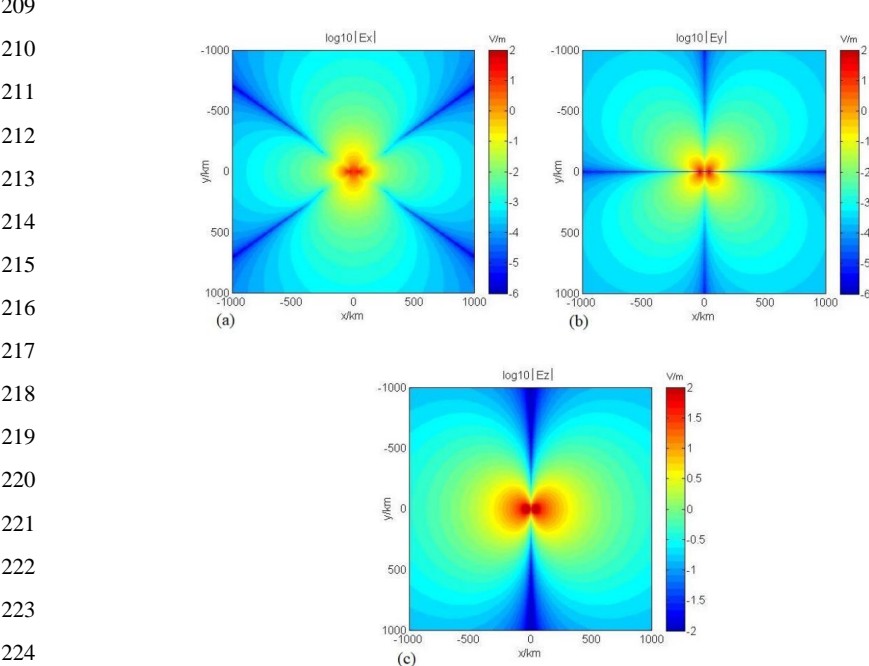










**Figure 2.** Two-dimensional distributions of electrical field components of (a) Ex, (b) Ey, and (c) Ez, respectively
induced by the Wenchuan seismo-current $I = 1.5 \times 10^5$ kA at $f = 0.01$Hz after making a logarithm calculation at the
Earth's surface.

Figure 2 presents the 2-D power radiation patterns of three electrical components $E_x$, $E_y$, and $E_z$,
respectively at the Earth's surface induced by the Wenchuan dipole source $I = 1.5 \times 10^5$ kA at $f = 0.01$
Hz within 1000 km from the epicenter. From Figure 2, we can see that the magnitude of the electrical
field intensity near the Wenchuan epicenter can be up to kV m$^{-1}$ for each component, for instance, $10^2$
kV m$^{-1}$ for $f = 1$ Hz and $10^4$ kV m$^{-1}$ for $f = 10$ Hz. From this point, we also can infer that the maximum
electrical field at the ground for all frequency band of $f = 0.01$–10 Hz during the Wenchuan event can
beyond kV m$^{-1}$ order. The maximum value central is 3.25 kV m$^{-1}$ for the vertical field $E_z$ at $f = 0.01$ Hz
near the center when the calculated electrical values have been examined, which will be as an input
into an electric field penetration model developed by Zhou et al. (2017) in the following part.

**3. Electrical coupling between the ground surface electromagnetic emissions and**
**ionosphere**
3.1. Basic equations and boundary conditions
The electric field penetration model developed by Zhou et al. (2017) has been briefly reviewed
first.
In the atmosphere, when the duration of a seismic event is equaling to or more than the
atmospheric electric field relaxation time $\tau_0$, the atmospheric condition can be transmitted from an
arbitrary initial state to a final steady state, which complies with the Ohm's law and the charge
conservation condition. Now, the electrostatic potential equation has been attained:



$$-\nabla \cdot (\bar{\bar{\sigma}} \cdot \nabla \Phi) = Q \qquad (1)$$
where $\bar{\bar{\sigma}}$ is the conductivity tensor.
In a Cartesian coordinate system, where, $z$-axis is vertically upward, $x$-axis is toward magnetic
south, and $y$-axis is toward magnetic east,
Equation (Eq.) (1) can be written as

$$\left(S^2 + \frac{\sigma_\parallel}{\sigma_P}C^2\right)\frac{\partial^2 \Phi}{\partial x^2} + \frac{\partial^2 \Phi}{\partial y^2} + \left(C^2 + \frac{\sigma_\parallel}{\sigma_P}S^2\right)\frac{\partial^2 \Phi}{\partial z^2} - 2\left(1 - \frac{\sigma_\parallel}{\sigma_P}\right)CS\frac{\partial^2 \Phi}{\partial x \partial z} - \frac{CS}{\sigma_P}\left(\frac{\partial \sigma_P}{\partial z} - \frac{\partial \sigma_\parallel}{\partial z}\right)\frac{\partial \Phi}{\partial x} - \frac{C}{\sigma_P}\frac{\partial \sigma_H}{\partial z}\frac{\partial \Phi}{\partial y} +$$
$$\frac{\partial(\sigma_P C^2 + \sigma_\parallel S^2)}{\sigma_P \partial z}\frac{\partial \Phi}{\partial z} = Q(x,y,z) \qquad (2)$$

where $\sigma_\parallel$, $\sigma_P$ and $\sigma_H$ are the parallel, Pedersen and Hall conductivity, respectively. Magnetic field
lines are in the $x$-$z$ plane. $I$ is the dip angle, $S = \sin I$, and $C = \cos I$. $Q(x,y,z)$ is the current source. In
this LAIC model, we consider that the potential source is $j_z = \sigma_0 E_{z=0}$, and here $E_{z=0}$ is the vertical
component of the electrical field induced at the Earth's surface by the Wenchuan seismo-elluric current
$I$ attained above and $Q(x,y,z) = 0$, which are treated as the lower boundary condition during the
following calculations. However, a direct current is expected when this LAI coupling model developed
by Zhou et al. (2017). Taking this limit under consideration, we have to assume that the ULF
electromagnetic emissions appeared during the Wenchuan event meet the like-steady condition at the
lower frequency band here. At the same time, Li et al. (2019) have qualitatively investigated the real
time recordings from ground-based electromagnetic observation, geomagnetic observation and
ionospheric observation occurred three days prior to the Wenchuan event. And the results show that
their evolutionary processes in time reached the climax simultaneously within dozens of hours on 9
May, which indicates the LAI coupling occurred.
For the upper boundary condition, $z_\infty$ has been set as the upper boundary of the ionosphere and
no current flows out of this boundary. As we all know, variations of conductivity cover a large sacle
form the Earth's surface to ionosphere. Thus, the conductivity profile has been divided into three parts:
I neutral atmospheric regionwithz < 50 km, here the conductivity is considered to be isotropic,
$\sigma_\parallel(0) = \sigma_P(0) = 1.0 \times 10^{-14}\,\mathrm{S\,m^{-1}}$ and $\sigma_\parallel(z) = \sigma_P(z) = \sigma(0)\exp\left(\frac{z}{6\mathrm{km}}\right)$.
II atmosphere-ionosphere transition area with 50 km < z < 90 km, wconsidering the continuity of
the conductivity profile, we adopt the following formula: $\sigma_\parallel(z) = \sigma_1 \exp\left(\frac{z-50}{h_\parallel}\right)$, and $\sigma_P(z) =$
$\sigma_1 \exp\left(\frac{z-50}{h_P}\right)$.
III ionosphere with $z > 90$ km, where the conductivity mainly depends on charged particles,
cyclotron frequency of electron and ion, and their collision frequency, $\sigma_\parallel = \sum_a \frac{e^2 n_a}{m_a v_a}$ , $\sigma_P =$
$\frac{1}{B}\sum_a \frac{v_a \Omega_a}{v_a^2 + \Omega_a^2}en_a$, and $\sigma_H = \frac{1}{B}\sum_a \frac{\Omega_a^2}{v_a^2 + \Omega_a^2}en_a$, where $a$ stands for the $a$ species charged particles and
$n_a, v_a$ and $\Omega_a$ are the number density, collision frequency and cyclotron frequency of the particles.
More details can be found in Zhou et al. (2017).

For high-latitude region, where the magnetic lines are vertical, the dip angle $I = 90\,^\circ$, $S = 1$ and $C$
$= 0$. Eq. (2) can be simply written as:

$$\sigma_\parallel \frac{\partial^2 \Phi}{\partial z^2} + \sigma_P \frac{\partial^2 \Phi}{\partial x^2} + \sigma_P \frac{\partial^2 \Phi}{\partial y^2} + \frac{\partial \sigma_\parallel}{\partial z}\frac{\partial \Phi}{\partial z} = 0 \qquad (3)$$

With the lower-boundary and the upper-boundary conditions considered, the current distribution in
the atmosphere will be attained by solving the electric potential Eq. (3), as well as in the ionosphere
with the constrain of the conductivity defined above.


The simulation results gotten by Zhou et al. (2017) at high latitude have shown that the vertical
current produced by the additional surface vertical electrical filed flows into the ionosphere without
losing in the neutral atmosphere and then this current could induce abnormal ionospheric electrical
field. This process at high latitude can also be suitable for low-mid latitude due to an exponential
decrease of atmospheric conductivity, which is not dependent on a background magnetic line.
The current form the atmosphere has been considered as the source term and the electrical
penetration model in the ionosphere can be solved referred to the propagation of thundercloud electric
fields into the ionosphere. Under these conditions when the magnetic line is not vertical and the Earth's
magnetic field is treated as a simple dipole field, Eq. (3) can be transformed into the form of the dipole
coordinate system:
$$\frac{\partial}{\partial t}\left(a\frac{\partial \Phi}{\partial t}\right) + \frac{\partial}{\partial \varphi}\left(b\frac{\partial \Phi}{\partial \varphi}\right) + \frac{\partial c}{\partial \varphi}\frac{\partial \Phi}{\partial t} - \frac{\partial c}{\partial t}\frac{\partial \Phi}{\partial \varphi} = -\frac{\partial}{\partial S}(h_\varphi h_t j_s) \qquad (4)$$

where, $a = -\sigma_P \frac{h_\varphi h_s}{h_t}$, $b = -\sigma_P \frac{h_t h_s}{h_\varphi}$, $c = -\sigma_h h_s$.
Considering the conjugated effect (the potentials of ionospheric south and north ends of a
magnetic line equals each other due to a high ionospheric conductivity), Eq. (4) can be integrated along
the magnetic field lines connecting the lower boundaries of the opposite hemispheres:
$$A\frac{\partial^2 \Phi}{\partial z^2} + B\frac{\partial^2 \Phi}{\partial \varphi^2} + D\frac{\partial \Phi}{\partial t} + E\frac{\partial \Phi}{\partial \varphi} = F \qquad (5)$$

where $A$, $B$, $C$, $D$, $E$ and $F$ are parameters concerned with $a$, $b$, $c$, $h_\varphi$, $h_t$  $h_s$ in equation (4), $S_1$ and
$S_2$ are the lower and upper boundaries of the northern hemisphere, $S_3$ and $S_4$ of the southern
hemisphere, respectively. The range of the conductive ionosphere has been set to 90–500 km. More
details can be seen in Zhou et al. (2017).
Eq. (5) is the ionospheric potential equation. Unlike Eq. (3), Eq. (5) is a two-dimensional
equation but with a nonhomogeneous term as the electrical field source propagating along *s* direction:
$j_{s1}$ and $j_{s2}$ in lower and upper northern ionospheric boundaries. When the region taken part in
calculation is far away from the pole and the magnetic equator, the solution area is $(t_{\min} \times t_{\max}) \times$
$(\varphi_{\min} \times \varphi_{\max})$, where $t_{max} = t_0 + 0.1$ and $t_{min} = t_0 - 0.1$ are the magnetic field lines close to and
far away from respectively the magnetic equator, and $\varphi_{min} = \varphi_0 - \frac{10}{360}\pi$ and $\varphi_{max} = \varphi_0 + \frac{10}{360}\pi$
are the left and right boundaries of the magnetic longitude. If this boundary is far away from the
areawhere the current flows, Dirichlet conditions are well met and the boundary potential can be
considered to 0:
$$\Phi(t_{min,max}, \varphi) = 0 \text{ and } \Phi(t, \varphi_{min,max}) = 0 \qquad (6)$$

Under these boundary conditions, the potential of the ionosphere in Eq. (5) is a 2D elliptic partial
differential equation and can be solved using a relaxation iterative method. Combined with Eq. (4) and
the solution of Eq. (5), the distribution of electric field in the ionosphere will be attained.

3.2. Propagation of additional current in atmosphere and ionosphere
The current density on the ground can be attained $j_z = \sigma_0 E_{z=0}$ when the conductivity
$\sigma_0 = 10^{-14}$ S m$^{-1}$, where $E_{z=0}$ is the calculated vertical electrical field by the Wenchuan source $I =$
$1.5 \times 10^8$ A at $f = 0.01$ Hz at the Earth surface and its radiation pattern could be refer to Figure 2c in
Section 2. Thus, we can attain that the maximum value of the current density near the Wenchuan
source at the Earth's surface $j_{\max(z=0)}$ is of $10^{-11}$ A m$^{-2}$. This current propagates upward continuously
among the atmosphere and the ionosphere. Equations established and related boundary conditions in
Section 3.1 have been utilized to simulate this process.



Figure 3 presents two-dimensional distributions of calculated electrical fields of (a) north
component $E_x$, (b) east component $E_y$, and (c) vertical component $E_z$, respectively in the magnetic x-y
plane at the ionospheric bottom with an altitude z = 90 km. It is apparently shown that the induced
electrical field with kV m$^{-1}$ at the ground surface (See Figure 2) has been subjected to a severe
attenuation in the atmosphere to 0.1mV m$^{-1}$ magnitude at the ionospheric bottom. Zhou et al. (2017)
have demonstrated that, compared with high latitude region, in the mid-low latitudinal regions, the
intensity of the total horizontal electric field increases with the latitude and the vertical electric field is
more evident at low latitudes
On one hand, this current propagates continuously with a rapid attenuation in the well-conductive
ionosphere. Figure 4 exhibits the vertical electrical field distribution of this current flow in x-z plane
with an increasing altitude under the constraint of conductivity defined above. As presented in Figure 4,
the vertical current splits from its source and transmits in the ionosphere continuously with a rapid
attenuation, especially for the height beyond 150 km and eventually vanishes at the top of the
ionosphere with z = 500 km. The penetration height of additional electric field in the ionosphere is
higher at high latitudes than that at low latitudes (Zhou et al., 2017). On the other hand, the additional
electrical current from the atmosphere modifies the ionospheric parameters either by heating the
ionosphere or by the plasma drifting in the electrical field of this current (Ruzhin et al., 2014). Thus,
the additional potential determined by this current will be considered as an input in the following part.

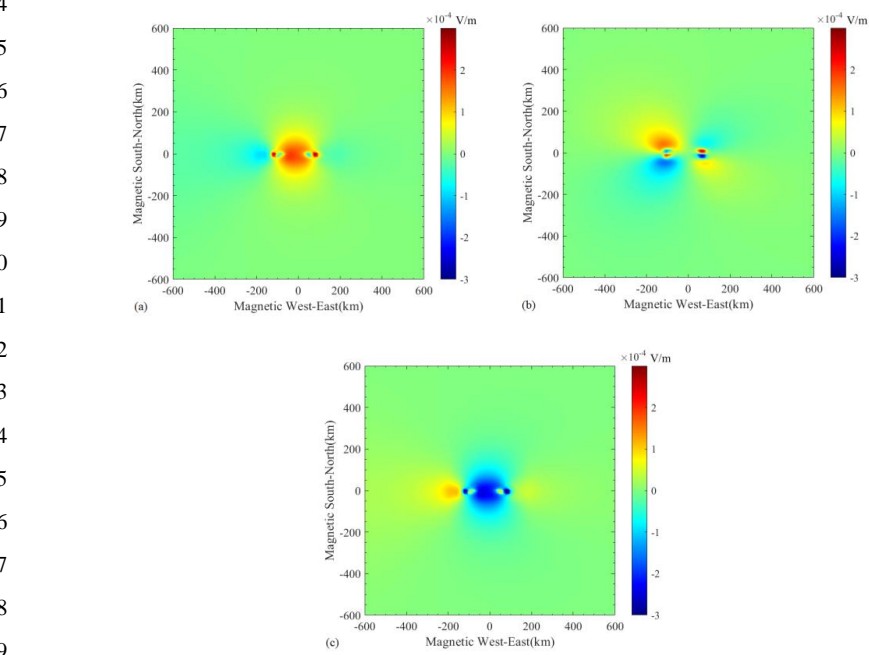

**Figure 3.** Two-dimensional distributions of the calculated electric field components at the ionospheric bottom (altitude 90 km) when the surface vertical current source of Figure 2c as an input into the atmosphere-ionosphere coupling model. This current source is the induction at the Earth surface from the Wenchuan finite length dipole at the operating frequency $f$ = 0.01 Hz. (a) Horizontal north field Ex; (b) Horizontal east field Ey; (c) Vertical field Ez.


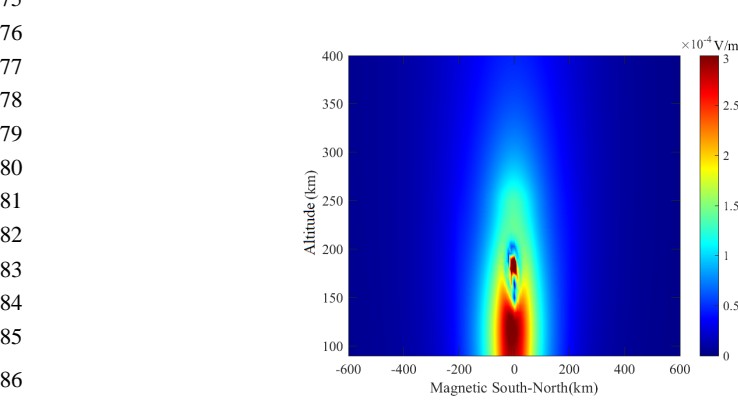

**Figure 4.** Distribution of the additional vertical electrical field caused by the Wenchuan source at the frequency $f = 0.01$ Hz at x-z plane in the ionosphere.

## 4. Ionospheric variations caused by the additional potential

Here, the TIE-GCM (Thermosphere-Ionosphere-Electrodynamics General Circulation Model) has been employed to attain ionospheric modifications caused by the Wenchuan source at the frequency $f = 0.01$ Hz. The TIE-GCM is a self-consistent physical model developed by National Center for Atmospheric Research (NCAR) and it is a comprehensive thermosphere and ionosphere coupling system to solve the three-dimensional momentum, energy and continuity equations for neutral and ion species using finite difference method. With polar particle deposition, high latitude electrical field and tidal effect from lower atmosphere considered, the TIE-GCM can calculates global distributions of the neutral gas temperature and winds, the height of the constant pressure surface and the number densities of the major constituents within the altitude 90–700 km when several parameters, such as F10.7 daily index, Kp index, etc. act as input (Roble et al., 1988; Richmond et al., 1992; Rougier et al., 2007). The additional potential induced by the Wenchuan source at the bottom of the ionosphere also as an input into this dynamic model after some modifications on its original codes.

Figure 5 demonstrates ionospheric influence with percentages of the additional potential on the electron density (Ne) at an altitude ~400 km (Figure 5a) and the total electron content (TEC) (Figure 5b). From Figure 5, it is clear that the additional potential from the Wenchuan source can causes ionospheric variations on both parameters in the Wenchuan epicentral area, as well as in its magnetically conjugated area, which is highly coincident with what have been described by most authors (Zhao et al., 2010; Liu et al., 2009; Pulinets et al., 2009; Yan et al., 2012; *and references therein*) in light of real-time ionosphere measurement. But the variation in this time is with a little magnitude of ~0.01%. Many documents have presented positive and negative variations on these both parameters at the opposite hemispheres prior to the Wenchuan EQ (Zhao et al., 2008; Yu et al., 2009; Liu et al., 2015; Akhoondzadeh et al., 2010; Pulinets & Ouzounov, 2011; *and references therein*). He et al. (2011) have reported a 30% increase of electron density during the Wenchuan event. Zeng et al. (2009) have presented more than 20% variations on electron density, electron temperature and $O^+$ density. It also has been reported 20%–60% negative and positive fluctuations on TEC prior to this event (Yan et al., 2012; Yu et al., 2009). Comparatively, the simulation results in this time are less far from real-time recordings by different sensors.

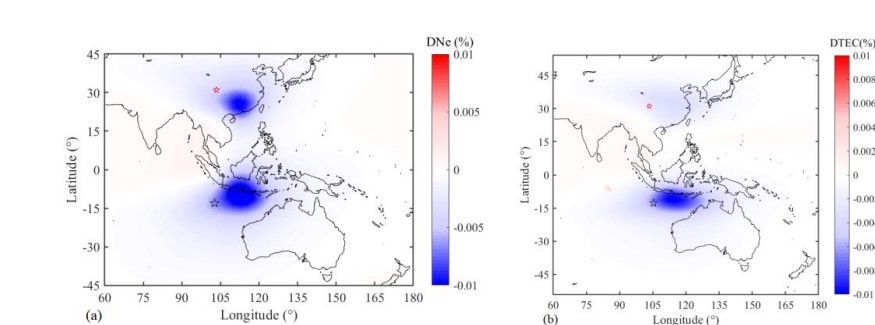

**Figure 5.** Ionospheric influence of the Wenchuan source at $f$ = 0.01 Hz on (a) electron density Ne and (b) the total electron content TEC. The Wenchuan epicenter and its magnetically conjugated point have been labeled by a red star and a black star, respectively in each panel.

## 5. Discussion

The fact that abundant electromagnetic emissions on ULF–ELF electrical field (Li et al., 2009, 2013; Gao et al., 2010; An et al., 2013; Jin et al., 2020), as well as ULF geomagnetic anomaly (Hu et al., 2009; Wang et al., 2009; Hayakawa et al., 2015; Li et al., 2015; Cheng et al., 2010; Li et al., 2019), before the 12 May 2008 Wenchuan $M_S$8.0 earthquake have been gradually reported. At the same time, ionospheric variations registered by different equipment of ground-based ionosonde data (Zhao et al., 2008; Sun et al., 2011; Maurya et al., 2013; *and references therein*), DEMETER satellite data (Zhang et al., 2009; Onishi et al., 2011; Liu et al., 2015; Walker et al., 2013; *and references therein*), ground-based GPS satellite data (Zhao et al., 2010; Liu et al., 2009; Pulinets et al., 2009; Zhu et al., 2009; *and references therein*), radio occultation data from six microsatellites of FORMOSAT3/COSMIC (F3/C) data (Liu et al., 2009; Ma et al., 2014; Hsiao et al., 2010), and CHAMP (challenging minisatellite payload) satellite data (Ryu et al., 2014) have also been confirmed as an increasing number of literatures published. These anomalies present different time scales and varied magnitudes but take on a common climax on May 9, 2008, three days prior to the Wenchuan main event, which undoubtedly raises an upsurge on theoretical or speculative interpreting them and investigating a probable LAI coupling concerned with all aspects on its mechanism.

Li et al. (2019) have qualitatively analyzed temporal variation orders of ground-based ultra-low frequency (ULF) electrical field, geomagnetic field and ionospheric parameters occurred on May 9 2008. They results indicate an LAIE coupling process: the electromagnetic energy propagates from the epicentral area to the Earth's surface, via the atmosphere and ionosphere, finally to its magnetically conjugated area in the opposite hemisphere, causing ground-based, atmospheric and ionospheric electromagnetic disturbances, in that order. In this research, theoretical simulations on this LAI electromagnetic coupling process have been performed on the basis of an observable 1.3 mV m$^{-1}$ ULF ($f$ = 0.01–10 Hz) electrical field registered at 1440 km Gaobeidian observing station in Hebei Province. An infinite length electrical dipole in half-space model has been utilized beneath the Earth to estimate the "energy source" of this LAI coupling and the calculated seismo-telluric current $I$ lies in the range of $1.5 \times 10^5$–$3.4 \times 10^6$ kA corresponding to the frequency range of $f$ = 0.01–10 Hz. Bortnik et al. (2010) have employed an electrical dipole collocated with the 31 October 2007 "Alum Rock" $M_W$ = 5.6 earthquake



463 hypocenter to interpret an observable 30 nT pulse at 1 Hz and D = 2 km and their results present an

464 estimated seismo-telluric current ~10–100 kA. Comparatively, the result attained in this time is

465 probably in a reasonable range (Li et al., 2016).

466  In the atmosphere, an electrical field penetration model developed by Zhou et al. (2017) has

467 been used to simulate the propagating properties of the surface current from the Wenchuan source.

468 Corresponding to the direct current in this model, the vertical electrical field $E_z$ produced by the

469 seismo-telluric current $I = 1.5 \times 10^5$ kA at $f = 0.01$ Hz has acted as the primary input into this model.

470 The central magnitude of this input electrical field on the ground surface can be up to kV m$^{-1}$ and this

471 field attenuates to the order of 0.1 mV m$^{-1}$ at the bottom (z = 90 km) of the ionosphere, which merely

472 leads to ~0.01% variations on electron density at the altitude 400 km and TEC in the ionosphere, an

473 incomparable value with real-time measured. Concerned with the electrical penetration model, Zhou et

474 al. (2017) have proposed that 1000 V m$^{-1}$ vertical electrical filed on the ground can generate $1.1 \times 10^{-6}$ V

475 m$^{-1}$ electrical field at the ionospheric bottom at the magnetic latitude of 30 °N, which is too smaller than

476 background ionosphere electrical filed of 1–3 mV m$^{-1}$ to generate ionospheric fluctuations by

477 electrodynamic processes. Therefore, a dramatic increase in atmospheric conductivity by neutral

478 atmosphere ionizing is suggested by Zhou et al. (2017).

479  Kuo et al. (2014) have presented a 20% ionospheric variation induced by 5 mV m$^{-1}$ electrical

480 field corresponding to 100 nA m$^{-2}$ current density at the ground surface. To get an obvious ionospheric

481 variations, by an LAI coupling model, Kim & Hegai (1999) and Pulinets et al. (2000) proposed that

482 obvious vertical electric fields at the Earth's surface could transmit into the ionosphere via the

483 atmosphere and modify dynamic and electronic properties of the ionosphere prior to the earthquake

484 Their results have presented that a ~1 kV m$^{-1}$ vertical electric field at the Earth's surface can be

485 expected to produce a ~1 mV m$^{-1}$ horizontal electric field at the ionospheric heights to give rise to

486 obvious ionospheric variations.

487  Sorokin & Hayakawa (2013, 2014) have presented that the injection of charged aerosols into

488 the atmosphere acts as an additional EMF (electro-motive force) in the global circuit. Under this

489 condition, the total electric current can be written as $\mathbf{j} = \sigma\mathbf{E} + \mathbf{j}e$, here $\mathbf{j}e$ is EMF external current and $\sigma\mathbf{E}$

490 is conductive current. The additional electrical filed can be up to 10 mV m$^{-1}$ leading to ionospheric

491 variations when the EMF external current acts as a compensate term during the conductive current

492 propagating up to the ionosphere. On the basis of model, Yang et al. (2014) have performed a

493 numerical simulation on seismic-related DC electric filed during atmosphere-ionosphere coupling.

494 Their results have presented that the horizontal scale of electric field in the ionosphere is larger than

495 that of the external current in the atmosphere and this current can induce much larger electric field at

496 night than during the day.

497  As a discussion, we have also performed all corresponding calculations on other observing

498 frequencies at $f = 1$ Hz and $f = 10$ Hz, respectively, like done on $f = 0.01$ Hz. The calculated

499 magnitudes of ground-surface vertical electrical fields near the central points for these two frequencies

500 can be up to $10^2$ kV m$^{-1}$ and $10^4$ kV m$^{-1}$, corresponding to respective ionospheric-bottom electrical

501 fields of ~200 mV m$^{-1}$ and ~1000 mV m$^{-1}$. Figure 6 shows differences of electron density (DNe) for

502 different layers defined in TIE-GCM model of 10 (~100 km in altitude), 30 (~200 km), 35 (~250 km),

503 40 (~300 km), 50 (~400 km) and 55 (~450 km) due to an input electrical potential determined by the

504 Wenchuan source at the frequency $f = 1$ Hz. As shown in Figure 6, the patterns and the magnitudes of

505 the DNe during this period are variable in different altitudes near the epicenter and its magnetically

506 conjugated area. It is shown clearly that the magnitude of positive variations increases as the altitude

increases below ~300 km and can reach 5% at 200–300 km, while the maximum varied magnitude can
be up to 10% but with negative anomaly at ~400 km, which is highly coincident with the work done by
He & Heki (2016, 2018). They have investigated the variations of the total electron content (TEC)
before two mid-latitude EQs of the 2010 Maule $M_W$8.8 EQ and the 2015 Illapel $M_W$8.3 EQ in Chile and
the results have demonstrated that the ionospheric anomalies have displayed both positive and negative
regions, with different altitudes of ~200 km and ~400 km, respectively, distributing roughly along the
geomagnetic fields.

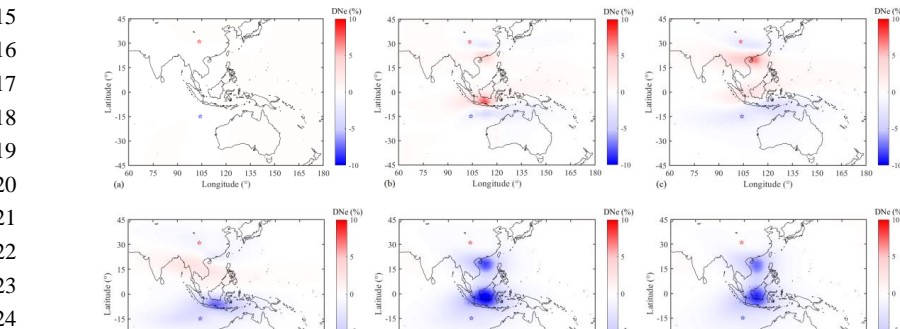

**Figure 6.** Ionospheric influence of the Wenchuan source at $f$ = 1 Hz on electron density Ne at different layers
defined by the TIE-GCM model. The Wenchuan epicenter and its magnetically conjugated point have been labeled
by a red star and a blue star, respectively in each panel. (a) layer 10 (~100 km in altitude), (b) 30 (~200 km), (c) 35
(~250 km), (d) 40 (~300 km), (e) 50 (~400 km) and (f) 55 (~450 km).
Figure 7 exhibits ionospheric influence with percentages of the additional potential on the electron
density (Ne) at an altitude ~400 km (Figure 5a) and the total electron content (TEC) (Figure 5b). From
Figure 7, it is clear that the additional potential from the Wenchuan source can cause negative
variations on both parameters in the Wenchuan epicentral area, as well as in its magnetically
conjugated area. The varied percentage could be up to 15%, and this value is also consistent with the
real-time measured magnitude.

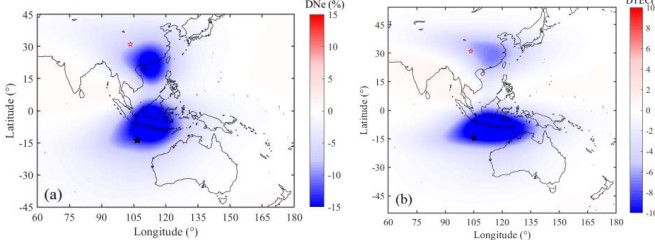

**Figure 7.** Ionospheric influence of the Wenchuan source at $f$ = 10 Hz on (a) electron density Ne at the high altitude
400 km and (b) the total electron content TEC. The Wenchuan epicenter and its magnetically conjugated point





have been labeled by a red star and a blue star, respectively in each panel.

Apparently, variations of ionospheric parameters seem to be approximately proportional to the
magnitude of the additional potential input. The varied magnitude on both parameters of electron
density and the total electron content is covering 0.01–15% corresponding to the observing frequency
band of $f$ = 0.01–10 Hz during the Wenchuan event. While this potential magnitude has been
determined by the frequency of the signals observed during the earthquakes. Concerned with
electromagnetic signals originated from seismic activities, on one hand, ULF signals can propagate to
the ground surface due to their lower attenuation beneath the Earth. Li et al., (2013) have reported
~0.1–0.3 s electrical signals recorded during the climax stage of the anomaly before the Wenchuan
main event. On the other hand, most experimental and real-time recordings have demonstrated that
signals with relatively higher frequencies appear during the main rupture of strong seismic events,
where the main rupture refers to a process of micro-cracks quickly developing into macro-rupture for
pressed fault (Li et al., 2013; Qian et al., 1996, 2003; Hao et al., 2003). Please note, this rupture
probably happens from the hypocentral depth till to the near Earth's surface during this period instead
of only focusing on the small area around the hypocenter of an impending earthquake. Therefore, even
signals with higher frequency than the ULF band can also easily propagate out of the Earth. Li et al.
(2013) have also noticed that the climax occurred on May 9, 2008, three days prior to the Wenchuan
main event, which indicates that an integrated effect of the electrical signals with different frequencies
acts as persisting current equivalent to a direct current to cause a LAI coupling.
A direct current electrical penetration model has been utilized here to simulate and interpret the
processing of the electrical current propagating in the atmosphere and ionosphere and simultaneously
causing ionospheric influence. It is noticed from the simulation results that the patterns of the
ionospheric variations are similar to what have been depicted in different literatures (Zhao et al., 2010;
Pulinets et al., 2010; Ryu et al., 2014; *and references therein*): the anomalies appear in bothside
hemispheres with a shift to the equator. Also, previous researches have shown that the location of
ionospheric effect is not coincident right with the vertical projection of the epicenter of the appending
earthquake but shifts equatorward at high and middle latitudes. Li et al., (2023) have reported statistical
seismo-ionospheric influence performed by electron density measured by the CSES (China Seismo–
Electromagnetic Satellite) for more than three years shifts 500–700 km away instead of right above the
epicenters for strong seismic activities in mid-low latitudes. Liu et al. (2009) have presented that the
eastward plasma E×B drift causes the GPS TEC enhancement slightly shifting to the east side of the
Wenchuan epicenter, although the generated mechanism is not understood. The enhanced GPS TEC
appeared in the southern China with a rounded shape, which is usually under control of the northern
equatorial ionization anomaly (EIA). So, bothside ionospheric anomalies generally shift toward the
equatorial area due to double crests of the EIA (Pulinets & Boyarchuk, 2004; Liu et al., 2009; Zhao et
al, 2008).
The ionospheric effect over the seismic activity areas in one hemisphere, as well as over their
magnetical conjugation areas in the opposite hemisphere could be depicted along the geomagnetic field
lines (Pulinets & Boyarchuk 2004). It is possible the space distribution of the TEC influence has been
determined by the joint effect of the two factors: the heating on the ionospheric properties by electric
current and the plasma drift in the electric field caused also by this current. The resulting E×B drift
should lead to a redistribution of plasmatic parameters over the earthquake preparation zone, as well as
over its magnetically conjugated area. The ionospheric anomalies in both sides of the sphere locate on




different sides of the magnetic meridian, which passes through the earthquake epicenter and its
magnetic conjugate point (Ruzhin, et al., 2014). However, the central points of the ionospheric
abnormities on both hemispheres in this time seem to locate the same side of the magnetic meridian
(see Figures 5–7). Also, noticed that, the spatial distribution and patterns of ionospheric variations
could change with the shape and direction of the underground source during the calculation. However,
we cannot determine the right shape of the real source causing by an earthquake. Another point is, the
atmosphere, as a common part of the half-space model and the electrical penetration model utilized in
this time, is of different properties: homogeneous medium with a specified conductivity in the first
model and inhomogeneous medium with various parallel conductivity $\sigma_\parallel$, Pedersen conductivity $\sigma_P$
and Hall conductivity $\sigma_H$ along the altitude in the second one, which could affect the final results. A
comprehensive model with more precise lithospheric and atmospheric conductivity information will be
expected to gain more reasonable results in the future.

### 6. Conclusions

This investigation aims to tentatively establish an LAI electromagnetic coupling process model in
the light of several physical models and real-time recordings of different spheres before the Wenchuan
$M_S$8.0 earthquake. A finite length electrical dipole in a half space model has been firstly employed to
estimate the possible magnitude of the "energy source" for an observable 1.3 mV m$^{-1}$ electrical field at
1440 km Gaobeidian station. The results show that the expected seismo-telluric current falls in the
range of $I = 1.5\times10^5$–$3.4\times10^6$ kA if observing ULF band of $f = 0.01$–10 Hz has been considered.
The electrical fields induced by this seismo-telluric current propagate from the Wenhucan hypocentral
area to the ground and the magnitudes of their vertical components are beyond kV m$^{-1}$ at the Earth's
surface, which will act as the input of the "energy source" arousing a coupling between the atmosphere
and the ionosphere.
Then, the electric field penetration model developed by Zhou et al., (2017) has been utilized here
to simulate the propagating process of the electrical field from the ground on one hand, and get the
additional electrical field at the bottom of the ionosphere at 90 km, on the other hand. It has been
testified that the magnitude of the additional electrical field at the bottom of the ionosphere ranges from
0.1 mV m$^{-1}$ to 1000 mV m$^{-1}$ if the observing frequency considered and then this field is subjected to a
severe decay in the well conductive ionosphere.
Finally, the magnitude of ionospheric variations induced by the additional field at the ionospheric
bottom has been simulated by the TIE-GCM and the expected results can be up to 15% if combined
effect of all recorded signals considered, which basically keeps the same magnitude as real-time
ionospheric measurements. However, much work, such as, selections of parameters, construction of a
comprehensive theoretical physical model, and so on, will be improved in the near future.

### Competing interests

The contact author has declared that none of the authors has any competing interests.

### Acknowledgments

This work was supported by the National Natural Science Foundation of China (NSFC) under Grant No. 41774084.
National Center for Atmospheric Research (NCAR) is supported by the National Science Foundation. The



TIEGCM model can be downloaded at http://www.hao.ucar.edu/modeling/tgcm/download.php.
**Data Availability Statement**
All data and code utilized in this manuscript can be downloaded at https://www.alipan.com/s/PRsZpAyoh3p.

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
