# Peer review of "Simulation of a lithosphere-atmosphere-ionosphere electromagnetic"

_Natural Hazards and Earth System Sciences, 2024_

## Referee Comment (RC2)

[referee-annotated manuscript omitted]

---

## Author Comment (AC1)

We thank the reviewers for their comments. Answers are given below in red. Changes in the revised version of the paper are also in red.

**Reviewer #3**

Dear authors and editor,

the manuscript "Simulation of a lithosphere-atmosphere-ionosphere electromagnetic coupling prior to the Wenchuan MS 8.0 earthquake" by Mei Li, Zhuangkai Wang, Chen Zhou, Handong Tan, and Meng Cao is an interesting research devoted to modeling the propagation of electro-magnetic anomalies produced by pre- and coseismic processes associated with the occurrence of the large Mw 7.9-8.2 2008 Sichuan event.

While the topic presents high scientific significance and the article is based on definitely good scientific research quality, I think that it suffers from a number of weaknesses preventing me to support its publication in NHESS if not after a major revision; however, the decision must go to the editor.

Below my major comments that I think should be addressed before the manuscript can be considered for publication:

1) The abstract should be a short, clear message summarising the content of the paper in a simple way that should be readable by non experts: addressed problem, state of the art, fundamentals of models, main idea, main result and its interpretation.

   Simple, direct: a take home message. Authors' abstract does not go to the point, it is complex, I cannot understand what you did in your study just reading it.

   We thank the reviewer for his comments on the abstract of the paper and we rewrite it completely in the revised version of the paper in red.

2) Several sentences in the main text are confusing, too long and contain language mistakes: I suggested several improvements in the attached pdf file to make the manuscript more enjoyable to readers.

   All mistakes labeled by the reviewer have been modified in red in the revised version of the paper.

3) Observations and simulations should not be confused: I think that a chapter devoted to show observations with maps etc should be added after the paragraphs devoted to models and then, discussions should be more focused on the

comparison of models and actual observations. Quantitative analysis to understand if the model works appropriately should be done (e.g., misfit assessment etc).

The main topic of this paper focuses on electromagnetic energy propagation from lithosphere (the hypocenter of the Wenhcuan EQ), via atmosphere, to the ionosphere to quantitatively establish a LAIEC model associated with the Wenchuan earthquake on the basis of real-time ULF electromagnetic observations (ground-based Gaobeidian station) and reported ionospheric observations prior to the Wenchuan event.

We have added some real-time recording pictures into Section 2 to display some related anomalous ULF electromagnetic information recorded prior to the Wenchuan earthquake. According to these recorded anomalous signals, the possible seismo-telluric current will be calculated using a finite-length electrical dipole in a half-space model. This current will act as the "energy source" to drive the total LAIEC process related to the Wenchuan earthquake. And then, the electric field penetration model and the TIE-GCM have also employed to simulate and calculate the propagating processes of the electromagnetic information in the atmosphere and ionosphere, as well as ionospheric variations caused by the additional electric field at the bottom of the ionosphere. During this period, many parameters have to be specified. However, giving accurate values to these parameters is difficult, which could lead to some uncertainties on our results. On this point, we have also discussed these uncertainties in Discussion section in the revised version of the paper.

4) It is not always clear which output of the model is compatible with physical phenomenons and which apparent signals are instead due to spurious effects, random fluctuations, computational instabilities (e.g., the red spot at the top of the two-peakes red anomaly in Figure 4 at about 180 km in height and 0 magnetic south-nord. Is it an artifact of the model?). Limits and hypotheses below the models should be sufficiently discussed.

It has been testified that the "hot-dot" in Figure 4 at about 180 km height is caused by the distortion in the central of the input ground surface electric source. The surface calculated electric values near the ground source central (the projection on the ground surface of the underground finite length dipole) are not accurate due to theoretical calculation method. We have added some comments in the reversion of the paper in red.

Minor comments, discretionary requests, suggestions and corrections of minor mistakes are listed throughout the attached pdf file for the sake of simplicity.

They have been corrected in the revised version of the paper.

Thanks for allowing me to review this work and for taking into account my humble comments.

We thank the reviewer for his careful comments. And further comments are welcome if it is necessary.

---

## Author Comment (AC2)

We thank the reviewer for the comments. Answers are given below in red. Changes in the revised version of the paper are also in red.

**Reviewer #1**

The paper makes a significant effort to simulate the electromagnetic coupling through the lithosphere, atmosphere, and ionosphere before the Wenchuan MS8.0 earthquake. The integration of geophysical observations with advanced modeling techniques is a noteworthy approach. Nevertheless, there are several issues that require the author's attention to enhance the robustness of the study.

1. In Section 2, the estimation of the current magnitude excited by the Wenchuan earthquake, derived solely from the seismic electromagnetic signals at the Gaobeidian station, may be overstated. The reliance on data from a single station may not provide a compelling argument; if the electromagnetic anomalies recorded at Gaobeidian are exceptional, closer stations should have reported stronger signals, yet such reports are absent. The authors should make some comments on this.

   The reviewer is right. Though electromagnetic emissions were recorded at three observing stations during the Wenchuan earthquake, only one ground-based observing station (Gaobeidian station) is taken part in calculations due to its almost synchronized anomalous emissions with the ionospheric variations three days prior to the Wenchuan main event. And it is 1440 km from the Wenchuan epicenter after all. Besides this, there are no near stations as reference at all. So, the calculation errors could be generated during this period. However, Guan et al. (1994) reported 16.9 mV m$^{-1}$ electric field at the 250 km Ningjin station in Hebei network before the Datong-yanggao $M_S$6.1 earthquake. The electric field of 1.3 mV/m at 1440 Gaobeidian station is a reasonable value if series attenuation of the wave along the distance is considered. We have added some discussion on this topic in the revised paper in Section 5.

2. The simulation's prediction of an induced electric field reaching up to 10^4 kV/m above the epicenter seems improbable. Is there any evidence from additional observational data or independent research supporting your simulation results?

   In the revised manuscript, we eliminate this value an only keep kV/m magnitude of the calculated ground electric field at f = 0.01 Hz. In fact, this value is a theoretical result but with an error near the source when the finite length dipole is used. We have added some discussion from different aspects in red to illustrate this unexpected strong result: ionospheric enhanced influence on remote propagating of incident wave, "selectivity effect" or

"sensitivity point" of electromagnetic signal measurement and calculated error of the physical model used in this paper.

3. The authors are requested to elucidate the cause of the small high-potential anomaly observed at an altitude of 150-200 km above the source, as depicted in Figure 4.

It has been testified that the "hot-dot" is caused by the distortion in the central of the input ground surface electric source. The surface theoretically calculated electric values near the ground source central are not accurate when the finite length dipole is used. We have mentioned that the central values of the electric fields on the ground near the Wenchuan source are not accurate due to theoretical calculation method.

4. At lines 461-465, the authors should clarify why the results from Bortnik (2010) are considered comparable with the findings of this study.

Generally, electromagnetic emissions are considered as the results of rock's deformation and rupture under the stress either from observing practice and rock-stress experiments. And the climax stage of these emissions occurs when the main rupture (quick development from the micro-cracks to macro-cracks) of the seismic fault under collective stress happens. The regional stress decrease is associated with the magnitude of the impending earthquake and released energy. Thus, electromagnetic emissions could be related with the magnitude or released energy of an earthquake. So, in the light of the formula between the released energy E and the magnitude M of an earthquake $lgE = 5.8 + 2.4M$ (Beno & Richter, 2010), the results attained in this paper are possibly in a reasonable range.

We have also added some similar contents into the revised version of the paper in red.

---

## Author Comment (AC3)

We thank the reviewer for the comments. Answers are given below in red. Changes in the revised version of the paper are also in red.

Reviewer #2

The paper deals with an interesting and challenging topic: the study of the lithosphere-atmosphere-ionosphere electromagnetic coupling before the occurrence of large earthquakes. Starting from the analysis of the ULF electromagnetic emissions observed by the Hebei geophysical network before the occurrence of the great Wenchuan earthquake (Ms8), the authors evaluated the magnitude of an energy source capable of generating the ULF signals observed by the Hebei network. In a second step, they simulated the propagation of the electrical signals through the atmosphere and finally obtained an estimate of the energy at the bottom of the ionospheric layer.

The overall organisation of the paper is quite good, but I have to make some critical comments. The main comments are as follows:

1. The estimate of the intensity of the "energy source" near the focus of the Wenchuan earthquake depends on the electromagnetic properties of the subsurface model (dielectric permittivity, magnetic permeability, conductivity). The assumption about the value of the conductivity could be better discussed, what are the changes if we modify the value of the conductivity? Furthermore, the assumption of a homogeneous half-space seems too simple, what are the possible changes introduced by a conductive shallow layer or by the presence of lateral discontinuities?

   The reviewer is right. Considered several electromagnetic properties like dielectric permittivity, magnetic permeability, conductivity, the conductivity is the predominant parameter that could affect the result much. In fact, we have discussed this topic in Li et al. (2016), when the results show that the observed electric field at 1440 km Gaobeidian station decreases about 20 orders of the magnitude if the conductivity of the Earth medium increases from $10^{-6}$ S m$^{-1}$ to 1 S m$^{-1}$. In this paper, we use a simple half-space model, considering the Earth medium is homogeneous, to estimate the possible seismo-telluric current acting as the driving source of LAIEC, which could lead to calculation errors. Not mention that the electromagnetic signals recorded before the Wenchuan earthquake are probably generated in the shallow layer of the Earth like what the reviewer has mentioned. So, in the next step, it is possible that a complex and comprehensive physical model, multilayer media model, for instance, will be developed.

   About this point, we have added some contents into Section 5 in red.

2. The paragraph 3.1 could be revised and reorganised. The model introduced by Zhou et al. (2017) has been applied to study the coupling between ground-based electromagnetic emissions and the ionosphere, but there are only purely qualitative considerations about the like-steady conditions of the electromagnetic emissions.

The model developed by Zhou et al. (2017) is suitable for like-steady conditions. It is shows that the switch time required to move from an arbitrary initial state to a final steady state is equal to or more than $\tau = 1000$ s. Li et al. (2019) have qualitatively analyzed the temporal variations of ground-based ULF electromagnetic emissions at Gaobeidian station, geomagnetic anomaly and ionospheric observations occurred on May 9, 2008, three days prior to the Wenchuan earthquake and found that this process lasted about 10 hours. And the electromagnetic emissions abruptly increased at 6:00–7:00AM and the ionospheric variations started at 1:00 PM and reached their climax at 4:00 PM. So, in this paper, we consider that the total coupling process complies with the like-steady conditions when the model is used. We have added similar contents into the revised version of the paper in red in Section 3.2.

3. The equations and mathematical formulae in paragraph 3.1 could be better described and simplified. This would make the paper more readable.

Yes. We have deleted some redundant contents about conductivities in different parts of the Earth's spheres, parameter details and boundary conditions for some equations.

4. In the paragraphs "5. Discussion" and "6. Conclusions", the novelty of the results and their implications could be better emphasized.

We have added some contents in red in the revise version of the paper.

Finally, there are some typewriting errors (see seismo-elluric) and some sentences that strongly require a re-formulation. An accurate revision of the English form is mandatory.

Thanks. We have modified some sentences and mistakes in red in the revised version of the paper.

---

## Author Comment (AC6)

We thank the reviewer for the comments. Answers are given below in red. Changes in the revised version of the paper are also in red.

Reviewer #3

This manuscript deals with modelling of low-frequency electric fields and currents caused by an underground current element in the lithosphere, atmosphere and ionosphere. This model is used in order to explain electric perturbation of about 1.3 mV/m                               observed                               during the Wenchuan earthquake at Gaobeidian Station at a distance of 1.440 km from the ep icenter of the earthquake.

The authors found that an underground source with a liner current of the order of $10^5$ - $10^6$ kA and a length of 150 km needs to produce this electric perturbation at such a great distance. The electric field on the earth's surface, calculated from this model, was used as an input parameter for another model describing the penetration of an electric field through the atmosphere into the ionosphere. The perturbation of the electric field in the ionosphere was shown to decrease to a value of 0.1 mV/m, while the TEC variations should be 0.01%.

The authors focuses on the electric field produced by the underground electric current. Meanwhile, this current produces not only an electric but also a magnetic field. Away from the currents source, this magnetic field can be roughly estimated using Bio-Savart law: $B \sim \mu_0 IL/(4\pi r^2)$. where $\mu_0$ is the magnetic constant; $I$ denotes the underground current; $L$ stands for the length of the current; and $r$ is the distance from the current element to the observation point.

Certainly, this law leaves out of account the influence of the boundary between the Earth and the ionosphere. Nevertheless, this law allows us to obtain an order-of-magnitude estimate. Substituting the author's parameters: $I$ = 1.5 $10^5$ - 3.4 $10^6$ kA and $L$ = 150 km as well the distance r = 100 km into the above equation, we obtain that $B$=(0.23 - 5.1) $10^{-3}$ T; that is, a value of one or two orders of magnitude greater than the Earth's magnetic field! At the distance $r$=1440 km (Wenchuan event) we obtain that B= (0.1 - 2.5) $10^{-5}$ T; that is, a value of the order of the Earth's magnetic field. Such strong magnetic perturbations never observed both before and after seismic events!

It seems likely that such a fantastically big value of the underground electric current is unrealistic. This means that the authors model cannot explain either electrical field registered at 1440 km Gaobeidian station during the Wenchuan earthquake or the ionospheric effects possibly related to this earthquake.

It makes no sense to dwell on another disadvantages of this model, since the drawback noted above is fatal. That is why I recommend, unfortunately, to reject this manuscript.

About produce mechanism of electromagnetic emissions before earthquakes, up to now, no clear explanation has been given although several physical mechanisms have been proposed to interpret the generation of EM emissions and electrical currents observed either during seismic activity or in the laboratory experiments. These include the electrokinetic and magnetohydrodynamic, piezomagnetism, stress-induced variations in crustal conductivity, microfracturing, etc. (Draganov et al.,1991; Park, 1996; Fenoglio et al., 1995; Egbert, 2002; Simpson and Taflove, 2005). Whatever the physical mechanism of electromagnetic generation is, it is well established that, during rock experiments conducted under laboratory conditions, a strong electrical current is produced when rocks are stressed, especially at the stage of the main rupture (Qian et al., 1996, 2003; Hao et al., 2003; Freund and Wengeler, 1982; Freund, 2002, 2009, 2010; Freund and Sornette, 2007; Scoville et al., 2015). So, to establish a physical or mathematical model is an effective way to interpret the observed electromagnetic emissions. In this research, we use a finite length current source beneath the Earth as an equivalent current induced by the Wenchuan event to interpret observed electric signal of 1.3 mV/m at 1440 km Gaobeidian station.

The reviewer mainly focused on the estimated current of $I = 1.5 \times 10^5$ - $3.4 \times 10^6$ kA and thinks that the induced magnetic field of B = $(0.1 - 2.5) \times 10^{-5}$ T in the light of Bio-Savart law is unreasonable.

Earth and Planetary Physics    doi: 10.26464/epp2019043    439

**Table 1.** Seismic P wave and geomagnetic disturbances

| Station | Epicentral distance (km) | First arrival of P wave (hh:mm:ss) | Time of M wave (hh:mm:ss) | Time difference of M and P wave (s) | D (°) | H (nT) | Z (nT) | F (nT) |
|---|---|---|---|---|---|---|---|---|
| | | | | | \multicolumn Amplitude of geomagnetic disturbance | | | |
| CD2 | 34 | 06:28:06 | 06:28:17 | 11 | 154.75 | 1044.86 | 983.76 | 16.11 |
| LZH | 565 | 06:29:19 | 06:30:33 | 74 | 2.59 | 8.32 | 9.22 | 0.32 |
| GYA | 597 | 06:29:21 | 06:29:32 | 11 | 13.03 | 87.7 | 89.56 | – |
| LCH | 644 | 06:29:13 | 06:29:47 | 34 | 40.24 | 239.33 | 317.49 | – |
| TOH | 769 | 06:29:41 | 06:31:31 | 110 | 0.37 | 0.63 | 0.97 | – |
| YCH | 875 | 06:29:58 | 06:31:21 | 83 | 113.96 | 993.2 | 665.46 | 0.87 |
| SHY* | 995 | | 06:30:06 | | 7.04 | 91.32 | 103.9 | 3.68 |
| GOM* | 995 | | 06:31:58 | | 1.19 | 0.97 | 0.52 | 0.2 |
| NNS | 1017 | 06:30:13 | 06:32:07 | 114 | 4.59 | 30.97 | 42.42 | – |
| JFE* | 1060 | | 06:33:32 | | 5.29 | 73.92 | 66.50 | – |
| JYG | 1086 | 06:30:26 | 06:33:34 | 188 | 0.73 | 0.89 | 0.63 | – |
| JIC | 1113 | 06:30:25 | 06:31:49 | 84 | 81.85 | 655.16 | 403.16 | – |
| WJH | 1218 | 06:30:38 | 06:31:57 | 79 | 53.84 | 372.63 | 241.98 | 0.85 |
| LSA | 1188 | 06:30:35 | 06:30:42 | 7 | 3.17 | 7.04 | 7.3 | – |
| TCH | 1462 | 06:31:07 | 06:31:13 | 6 | 6.57 | 54.33 | 49.28 | – |
| QZN | 1480 | 06:31:08 | 06:34:00 | 172 | 1.49 | 12.17 | 25.36 | – |
| JHA* | 1506 | | 06:34:54 | | 6.43 | 14.29 | 9.79 | – |
| BBS | 1523 | 06:31:16 | 06:31:20 | 4 | 7.66 | 110.52 | 62.91 | – |
| HAZ | 1598 | 06:31:26 | 06:31:28 | 2 | 6.31 | 73.73 | 82.81 | – |
| QZH | 1633 | 06:31:27 | 06:31:35 | 8 | 1.16 | 0.77 | 0.44 | – |
| DL2 | 1872 | 06:31:59 | 06:32:01 | 2.5 | 2.42 | 21.38 | 13.76 | – |
| HTB | 2070 | 06:32:21 | 06:39:04 | 403 | 0.36 | 1.07 | 0.57 | – |
| SQH* | 2206 | | 06:39:26 | | 0.17 | – | – | – |
| SGA* | 2372 | | 06:32:58 | | 4.48 | 42.95 | 30.16 | – |
| MZL | 2374 | 06:32:48 | 06:33:00 | 12 | 6.31 | 36.35 | 13.99 | – |
| KSH | 2666 | | | | – | – | – | – |
| DED | 2740 | 06:33:21 | 06:33:26 | 5 | 8.67 | 63.68 | 24.58 | – |

Note: * indicates we do not have seismic data from these observatories; '–' means no apparent disturbance was detected.

However, on one hand, we have already given some explanations on the attained look-like large current in the revised paper in red in Section 5 Discussion and please refer to that. On the other hand, Wang et al. [2019] have reported 1113 km JIC 403.16

nT, and 1218 km WJH 241.98 nT, coseismic geomagnetic disturbances in vertical Z component, while 1506 km JHA only 9.70 nT and 1523 km BBS 62.91 nT (see above table), and all these station are near and around Gaobeidian station used in our research. These observing stations show their apparent "selectivity" or "sensitivity" to signals. Compared with these really recorded measurements, the values we have attained in this paper seem reasonable. Of course, we can not use a physical model to comply with all these recordings. We also attached this reference:

Wang, Y. L., Xie, T., An, Y. R., Yue, C., Wang, J. Y., Yu, C., Yao, L., and Lu, J. (2019). Characteristics of the coseismic geomagnetic disturbances recorded during the 2008 $M$w 7.9 Wenchuan Earthquake and two unexplained problems. Earth Planet. Phys., 3(5), 435–443. http://doi.org/10.26464/epp2019043.

At the same time, even this large "energy source", the calculated ionospheric variation in the paper is still too low to compared with reported ionospheric varied magnitudes really measured by GPS TEC or other satellites three days before the Wenchuan event at observing $f = 0.01$ Hz.

However, please note, also in Discussion part of the paper, in the light of observing frequency $f = 0.01$–10 Hz, we have attained 10%–15% variations on ionospheric parameters at $f = 10$ Hz, and this value keeps the same order as really reported 20%–60% ionospheric variations during the Wenchuan event. We have emphasized two points, one is the recorded signals at Gaobeidian station maybe are with combined frequencies instead of single frequency. Another is, these signals were recorded just 3 days prior to the Wenchuan event, when it is possible that rupture occurred not only in the small area around the hypocenter but also in shallower layer, even near the ground area, so high frequency signals can be easily recorded at that time. Therefore, 10%–15% ionospheric variations are reasonable, although uncertainties still exist.

We advise that the reviewer read the revised version of the paper.

---

## Author Comment (AC8)

We thank the reviewer for the comments. Answers are given below in red. Changes in the revised version of the paper are also in red.

Reviewer #3

This manuscript deals with modelling of low-frequency electric fields and currents caused by an underground current element in the lithosphere, atmosphere and ionosphere. This model is used in order to explain electric perturbation of about 1.3 mV/m                                          observed                                          during the Wenchuan earthquake at Gaobeidian Station at a distance of 1.440 km from the ep icenter of the earthquake.

The authors found that an underground source with a liner current of the order of $10^5$ - $10^6$ kA and a length of 150 km needs to produce this electric perturbation at such a great distance. The electric field on the earth's surface, calculated from this model, was used as an input parameter for another model describing the penetration of an electric field through the atmosphere into the ionosphere. The perturbation of the electric field in the ionosphere was shown to decrease to a value of 0.1 mV/m, while the TEC variations should be 0.01%.

The authors focuses on the electric field produced by the underground electric current. Meanwhile, this current produces not only an electric but also a magnetic field. Away from the currents source, this magnetic field can be roughly estimated using Bio-Savart law: $B \sim \mu_0 IL/(4\pi r^2)$. where $\mu_0$ is the magnetic constant; $I$ denotes the underground current; $L$ stands for the length of the current; and $r$ is the distance from the current element to the observation point.

Certainly, this law leaves out of account the influence of the boundary between the Earth and the ionosphere. Nevertheless, this law allows us to obtain an order-of-magnitude estimate. Substituting the author's parameters: $I = 1.5 \ 10^5$ - $3.4 \ 10^6$ kA and $L = 150$ km as well the distance r = 100 km into the above equation, we obtain that $B=(0.23 - 5.1) \ 10^{-3}$ T; that is, a value of one or two orders of magnitude greater than the Earth's magnetic field! At the distance $r=1440$ km (Wenchuan event) we obtain that B= $(0.1 - 2.5) \ 10^{-5}$ T; that is, a value of the order of the Earth's magnetic field. Such strong magnetic perturbations never observed both before and after seismic events!

It seems likely that such a fantastically big value of the underground electric current is unrealistic. This means that the authors model cannot explain either electrical field registered at 1440 km Gaobeidian station during the Wenchuan earthquake or the ionospheric effects possibly related to this earthquake.

It makes no sense to dwell on another disadvantages of this model, since the drawback noted above is fatal. That is why I recommend, unfortunately, to reject this manuscript.

About produce mechanism of electromagnetic emissions before earthquakes, up to now, no clear explanation has been given although several physical mechanisms have been proposed to interpret the generation of EM emissions and electrical currents observed either during seismic activity or in the laboratory experiments. These include the electrokinetic and magnetohydrodynamic, piezomagnetism, stress-induced variations in crustal conductivity, microfracturing, etc. (Draganov et al.,1991; Park, 1996; Fenoglio et al., 1995; Egbert, 2002; Simpson and Taflove, 2005). Whatever the physical mechanism of electromagnetic generation is, it is well established that, during rock experiments conducted under laboratory conditions, a strong electrical current is produced when rocks are stressed, especially at the stage of the main rupture (Qian et al., 1996, 2003; Hao et al., 2003; Freund and Wengeler, 1982; Freund, 2002, 2009, 2010; Freund and Sornette, 2007; Scoville et al., 2015). So, to establish a physical or mathematical model is an effective way to interpret the observed electromagnetic emissions. In this research, we use a finite length current source beneath the Earth as an equivalent current induced by the Wenchuan event to interpret observed electric signal of 1.3 mV/m at 1440 km Gaobeidian station.

The reviewer mainly focused on the estimated current of $I = 1.5 \; 10^5$ - $3.4 \; 10^6$ kA and thinks that the induced magnetic field of B = (0.1 - 2.5) $10^{-5}$ T in the light of Bio-Savart law is unreasonable.

Earth and Planetary Physics    doi: 10.26464/epp2019043    439

**Table 1.** Seismic P wave and geomagnetic disturbances

| Station | Epicentral distance (km) | First arrival of P wave (hh:mm:ss) | Time of M wave (hh:mm:ss) | Time difference of M and P wave (s) | Amplitude of geomagnetic disturbance | | | |
|---|---|---|---|---|---|---|---|---|
| | | | | | D (') | H (nT) | Z (nT) | F (nT) |
| CD2 | 34 | 06:28:06 | 06:28:17 | 11 | 154.75 | 1044.86 | 983.76 | 16.11 |
| LZH | 565 | 06:29:19 | 06:30:33 | 74 | 2.59 | 8.32 | 9.22 | 0.32 |
| GYA | 597 | 06:29:21 | 06:29:32 | 11 | 13.03 | 87.7 | 89.56 | – |
| LCH | 644 | 06:29:13 | 06:29:47 | 34 | 40.24 | 239.33 | 317.49 | – |
| TOH | 769 | 06:29:41 | 06:31:31 | 110 | 0.37 | 0.63 | 0.97 | – |
| YCH | 875 | 06:29:58 | 06:31:21 | 83 | 113.96 | 993.2 | 665.46 | 0.87 |
| SHY* | 995 | | 06:30:06 | | 7.04 | 91.32 | 103.9 | 3.68 |
| GOM* | 995 | | 06:31:58 | | 1.19 | 0.97 | 0.52 | 0.2 |
| NNS | 1017 | 06:30:13 | 06:32:07 | 114 | 4.59 | 30.97 | 42.42 | – |
| JFE* | 1060 | | 06:33:32 | | 5.29 | 73.92 | 66.50 | – |
| JYG | 1086 | 06:30:26 | 06:33:34 | 188 | 0.73 | 0.89 | 0.63 | – |
| JIC | 1113 | 06:30:25 | 06:31:49 | 84 | 81.85 | 655.16 | 403.16 | – |
| WJH | 1218 | 06:30:38 | 06:31:57 | 79 | 53.84 | 372.63 | 241.98 | 0.85 |
| LSA | 1188 | 06:30:35 | 06:30:42 | 7 | 3.17 | 7.04 | 7.3 | – |
| TCH | 1462 | 06:31:07 | 06:31:13 | 6 | 6.57 | 54.33 | 49.28 | – |
| QZN | 1480 | 06:31:08 | 06:34:00 | 172 | 1.49 | 12.17 | 25.36 | – |
| JHA* | 1506 | | 06:34:54 | | 6.43 | 14.29 | 9.79 | – |
| BBS | 1523 | 06:31:16 | 06:31:20 | 4 | 7.66 | 110.52 | 62.91 | – |
| HAZ | 1598 | 06:31:26 | 06:31:28 | 2 | 6.31 | 73.73 | 82.81 | – |
| QZH | 1633 | 06:31:27 | 06:31:35 | 8 | 1.16 | 0.77 | 0.44 | – |
| DL2 | 1872 | 06:31:59 | 06:32:01 | 2.5 | 2.42 | 21.38 | 13.76 | – |
| HTB | 2070 | 06:32:21 | 06:39:04 | 403 | 0.36 | 1.07 | 0.57 | – |
| SQH* | 2206 | | 06:39:26 | | 0.17 | – | – | – |
| SGA* | 2372 | | 06:32:58 | | 4.48 | 42.95 | 30.16 | – |
| MZL | 2374 | 06:32:48 | 06:33:00 | 12 | 6.31 | 36.35 | 13.99 | – |
| KSH | 2666 | | | | – | – | – | – |
| DED | 2740 | 06:33:21 | 06:33:26 | 5 | 8.67 | 63.68 | 24.58 | – |

Note: * indicates we do not have seismic data from these observatories; '–' means no apparent disturbance was detected.

However, on one hand, we have already given some explanations on the attained look-like large current in the revised paper in red in Section 5 Discussion:

The first one is that, only one ground-based observing station (Gaobeidian station) is taken part

in calculations due to its almost synchronized anomalous emissions with the ionospheric variations three days prior to the Wenchuan main event. And it is 1440 km from the Wenchuan epicenter after all. Besides this, there are no near stations as reference at all. Guan et al. (1994) reported 16.9 mV m$^{-1}$ electric field at the Ningjin station in Hebei network before the 250 km Datong-yanggao $M_S$6.1 earthquake.

Secondly, electromagnetic signals associated with strong seismic activities are generally characterized by "selectivity" or "sensitive point". The selectivity effect is a complex phenomenon that may be attributed to a superposition of the following three factors: "source characteristics", "travel path" and "inhomogeneities close to the station" (Varotsos and Lazaridou, 1991; Varotsos et al., 2005). In the half-space model employed in this paper, the Earth medium is considered as homogeneous with unique conductivity $\sigma_1 = 1.0 \times 10^{-3}$ S m$^{-1}$. However, it is the fact that the Earth is inhomogeneous and even lateral discontinuous. Li et al. (2016) have reported that that the observed electric field at 1440 km Gaobeidian station decreases about 20 orders of the magnitude if the conductivity of the Earth medium increases from 10$^{-6}$ S m$^{-1}$ to 1 S m$^{-1}$. Li et al. (2019) have also reported that obvious geomagnetic anomaly on 8 May and 9 May 2008 in the same area as Gaobeidian station.

In addition, in the controlled-source electromagnetic (CSEM) method, widely used in petroleum exploration or mining, the ionospheric influence on EM fields should be considered when the distance between a large-scale and large-power fixed source and the receiver is up to 1000 km. Li et al. (2016) have attained that the ionosphere can prevent attenuating of the incident wave and the ionospheric influence on electromagnetic recordings at Gaobeidian distance can be up to 1–2 orders, which indicates that the 1.3 mV m$^{-1}$ electric field at this station considered in this research could be an enhanced value.

On the other hand, Wang et al. [2019] have reported 1113 km JIC 403.16 nT, and 1218 km WJH 241.98 nT, coseismic geomagnetic disturbances in vertical Z component, while 1506 km JHA only 9.70 nT and 1523 km BBS 62.91 nT (see above table), and all these station are near and around Gaobeidian station used in our research. These observing stations show their apparent "selectivity" or "sensitivity" to signals. Compared with these really recorded measurements, the values we have attained in this paper seem reasonable. Of course, we can not use a physical model to comply with all these recordings. We also attached this reference:

Wang, Y. L., Xie, T., An, Y. R., Yue, C., Wang, J. Y., Yu, C., Yao, L., and Lu, J. (2019). Characteristics of the coseismic geomagnetic disturbances recorded during the 2008 $M$w 7.9 Wenchuan Earthquake and two unexplained problems. Earth Planet. Phys., 3(5), 435–443. http://doi.org/10.26464/epp2019043.

At the same time, even this large "energy source", the calculated ionospheric variation in the paper is still too low to compared with reported ionospheric varied magnitudes really measured by GPS TEC or other satellites three days before the Wenchuan event at observing $f = 0.01$ Hz.

However, please note, also in Discussion part of the paper, in the light of observing frequency $f = 0.01$–10 Hz, we have attained 10%–15% variations on ionospheric

parameters at $f = 10$ Hz, and this value keeps the same order as really reported 20%–60% ionospheric variations during the Wenchuan event. We have emphasized two points, one is the recorded signals at Gaobeidian station maybe are with combined frequencies instead of single frequency. Another is, these signals were recorded just 3 days prior to the Wenchuan event, when it is possible that rupture occurred not only in the small area around the hypocenter but also in shallower layer, even near the ground area, so high frequency signals can be easily recorded at that time. Therefore, 10%–15% ionospheric variations are reasonable, although uncertainties still exist.

We advise that the reviewer read the revised version of the paper.